# A Phase-Adjustable Noise-Shaping SAR ADC for Mitigating Parasitic Capacitance Effects from PIP Capacitors

**DOI:** 10.3390/s25196029

**Published:** 2025-10-01

**Authors:** Xuelong Ouyang, Hua Kuang, Dalin Kong, Zhengxi Cheng, Honghui Yuan

**Affiliations:** 1National Key Laboratory of Infrared Detection Technologies, Shanghai Institute of Technical Physics, Chinese Academy of Sciences, Shanghai 200083, China; ouyangxuelong22@mails.ucas.ac.cn (X.O.); kuanghua22@mails.ucas.ac.cn (H.K.); kongdalin21@mails.ucas.ac.cn (D.K.); czx@mail.sitp.ac.cn (Z.C.); 2University of Chinese Academy of Sciences, Beijing 100049, China

**Keywords:** SAR ADC, NS, mode-switching, CMOS image sensor, parasitic capacitance

## Abstract

High parasitic capacitance from poly-insulator-poly capacitors in complementary metal oxide semiconductor (CMOS) processes presents a major bottleneck to achieving high-resolution successive approximation register (SAR) analog-to-digital converters (ADCs) in imaging systems. This study proposes a Phase-Adjustable SAR ADC that addresses this limitation through a reconfigurable architecture. The design utilizes a phase-adjustable logic unit to switch between a conventional SAR mode for high-speed operation and a noise-shaping (NS) SAR mode for high-resolution conversion, actively suppressing in-band quantization noise. An improved SAR logic unit facilitates the insertion of an adjustable phase while concurrently achieving an 86% area reduction in the core logic block. A prototype was fabricated and measured in a 0.35-µm CMOS process. In conventional mode, the ADC achieved a 7.69-bit effective number of bits at 2 MS/s. By activating the noise-shaping circuitry, performance was significantly enhanced to an 11.06-bit resolution, corresponding to a signal-to-noise-and-distortion ratio (SNDR) of 68.3 dB, at a 125 kS/s sampling rate. The results demonstrate that the proposed architecture effectively leverages the trade-off between speed and accuracy, providing a practical method for realizing high-performance ADCs despite the inherent limitations of non-ideal passive components.

## 1. Introduction

The analog-to-digital converter (ADC) is a crucial building block in modern optoelectronic sensor systems, responsible for bridging the analog world and the digital processing domain [1]. Within the diverse landscape of ADC architectures, the successive approximation register (SAR) ADC has gained prominence, particularly in energy-constrained applications, owing to its compelling advantages in structural simplicity, speed, and power efficiency [2]. This architecture, comprising a sample-and-hold circuit, a comparator, a digital-to-analog converter (DAC), and control logic, is versatile enough to support the digitization requirements of large- and small-pixel complementary metal oxide semiconductor (CMOS) image sensors (CIS).

While the charge-redistribution SAR ADC has dominated mainstream design over the past decade due to its excellent linearity and minimal static power consumption [3,4], specific applications impose unique constraints. For example, CMOS circuits designed for harsh environments such as cryogenic or high-radiation settings often rely on more robust, larger-node fabrication processes. In particular, poly-insulator-poly (PIP) capacitors emerge as the most suitable option, primarily due to their high reliability and widespread availability in standard fabrication processes. However, a principal drawback of these components is their significant parasitic capacitance. This non-ideality introduces harmonic distortion, which can become a performance bottleneck and degrade the overall accuracy of the ADC.

The work presented in [5] indicates that parasitic capacitance does not inherently limit the dynamic performance of an SAR ADC. It posits that the magnitude of parasitic-induced error, when quantified in least significant bits (LSBs), is independent of the ADC’s overall resolution. This principle, however, applies only to specific ADC architectures where the comparator is connected to the top plate of the capacitive array. In such configurations, bottom-plate parasitic capacitance does not affect the charge-sharing process during SAR conversion, whereas the top-plate parasitic capacitance merely induces minor signal gain attenuation.

## 2. Analysis of Parasitic Effects

This analysis begins with the ideal case, where the top plate of the capacitor array is connected directly to the comparator. To illustrate, a transition of the most significant bit (MSB) from logic low to logic high is considered. The corresponding change in the output voltage of the DAC, denoted as $\Delta V_{DAC}$, is given by the following:(1)$$\Delta V_{DAC}=\frac{C_{MSB}}{C_{total}}V_{Ref}$$

When the parasitic capacitance at the comparator’s input node ($C_{parasitic}$) is taken into account, this expression is transformed into the following:(2)$$\Delta V_{DAC}=\frac{C_{MSB}}{C_{total}+C_{parasitic}}V_{Ref}$$

A comparison between Equations (Equation 1) and (Equation 2) reveals that the primary effect of parasitic capacitance is signal attenuation, which manifests as a gain error in the ADC’s transfer function. Importantly, because this attenuation scales all conversion steps uniformly, the intrinsic linearity of the digital-to-analog conversion remains intact.

However, from the perspective of the SAR logic module, this gain attenuation prevents the DAC from spanning its intended analog voltage range, even when the maximum digital input code is applied. As a result, input voltages approaching full scale yield identical output codes, producing a subtle “clipping” effect, as illustrated in the transfer curve of Figure 1. This non-linearity is a key contributor to the emergence of odd harmonics observed in the output spectrum, a behavior demonstrated by the MATLAB (2024b) based simulation results for the Fast Fourier Transform (FFT) shown in Figure 2. In essence, the parasitic capacitance reduces the effective input swing, causing full-scale-induced precision degradation to occur several hundred millivolts earlier than expected.

To elucidate the impact of parasitic capacitance on the performance evolution of the ADC, a simulation was conducted.

In this analysis, the parasitic capacitance was swept across a range from 0.1% to 10% of the sampling-and-hold capacitance. The simulation results, presented in Figure 3, indicate a critical threshold. At a parasitic capacitance of 0.1% of the sampling-and-hold capacitance, the performance is limited by thermal noise, with the ADC achieving a high SFDR of approximately 80 dB. However, as this capacitance increases to 10%, the dominant source of performance degradation transitions to harmonic distortion. At this point, the energy from harmonic distortion accounts for approximately 97.8% of the total error budget, while the contribution from thermal noise drops to just 2.2%. This fundamental shift is reflected in the SFDR degrading significantly to 30.5 dB, a total drop of 49.5 dB.

To mitigate the performance degradation induced by parasitic capacitance, various techniques have been proposed in the literature. These approaches can be broadly classified into two categories. The first category encompasses physical design methodologies [6], which utilize optimized layout and routing strategies to match parasitic capacitances across the capacitor array systematically. The second category focuses on digital calibration techniques [7], in which specific node voltages are measured to estimate parasitic values. These estimates are then used to generate compensation data to correct the resulting errors.

More recently, noise-shaping techniques have gained prominence as a means to enhance the precision of SAR ADCs [8,9,10]. This approach is motivated by the observation that although SAR ADCs can operate at very high sampling rates, the target system—such as a CIS—often requires considerably lower data throughput. This disparity enables designers to trade excess sampling bandwidth for improved resolution via noise shaping, thereby boosting overall system performance.

This work presents a speed-adjustable SAR ADC architecture integrated with a phase control circuit. A noise-shaping mode can be activated by strategically inserting additional clock cycles, thereby enhancing the ADC performance. The investigation focuses primarily on the performance gains that noise-shaping techniques can provide for column-level ADCs within CMOS image sensors. The remainder of this paper is organized as follows. Section 2 presents the theoretical analysis and prototype design of the individual components of the proposed SAR ADC. Section 3 describes the complete circuit architecture and reports its measured performance. Finally, Section 4 concludes the paper with a summary of the results and analysis.

## 3. Design of a Column ADC Prototype

The architecture of the proposed noise-shaping SAR (NS-SAR) ADC is illustrated in Figure 4. The black-shaded portion of the diagram represents a conventional SAR ADC, comprising a DAC, SAR logic, comparator, and bootstrapped sampling switch. The red-shaded section highlights the feedback network that processes the DAC’s residual voltage and simultaneously functions as a loop filter to shape quantization noise.

The design of the loop filter is central to implementing noise shaping in a SAR ADC. Noise shaping in SAR ADCs is typically realized using one of two common topologies [11]: passive noise shaping, which primarily utilizes switched capacitors, and active noise shaping, which relies on operational amplifiers. In this work, a passive topology is adopted to adhere to the stringent power constraints of CMOS image sensors, as it incurs negligible additional power consumption.

The circuit is designed for fabrication using the AMS C35B4OA 0.35 µm process, a technology that facilitates monolithic integration with the photosensor on the same silicon die. The front-end photodetector employs a four-transistor (4T) pixel architecture [12], producing an input signal swing between 1 V and 2 V [13]. Consequently, the ADC is designed to operate from a 3.3 V supply.

### 3.1. Bootstrapped Sample Switch

The proposed SAR ADC uses a bootstrapped switch [14] to sample and hold the input voltage onto the top plate of the capacitor array. Once the switch turns off, the circuit transitions from the sampling phase to the hold phase, during which the SAR logic executes the conversion process. Although the sampling switch remains off during this phase, a time-varying input signal can still couple through the parasitic source-drain capacitance of the switch transistor and onto the DAC. This signal feedthrough introduces non-linearity into the final output of the ADC.

To mitigate this effect, a dummy switch technique is used, as illustrated in Figure 5. A dummy transistor, sized identically to the main sampling switch, is placed in parallel with the primary switch. Furthermore, meticulous attention is paid to the physical layout to ensure that both transistors reside in symmetrical and closely matched environments. The complementary differential input signal is applied to the dummy switch, establishing a cancellation path that significantly attenuates the undesired coupling from the input to the DAC. This approach enhances the overall sampling accuracy, as demonstrated in Figure 6.

To verify the effectiveness of the dummy switch, transient simulations of the sample-and-hold circuit were conducted. A 3.87 MHz sinusoidal input signal was sampled using a 32 MHz clock. Figure 7 presents the FFT spectra of the simulated results, comparing the circuit’s performance with and without the dummy switch.

Without the dummy switch, a prominent harmonic spur appears at 11.62 MHz. Incorporating the dummy switch suppresses this dominant harmonic by 17.5 dB, thereby improving the overall SNDR from 87.4 dB to 104.46 dB.

### 3.2. Comparator

In SAR ADCs, comparator design must balance high speed, low power consumption, and low noise. For CMOS image sensor applications, speed requirements are typically relaxed in favor of minimizing power and noise, making moderate-speed operation a viable option. While the StrongArm [15] latch is widely adopted, its inherent noise and offset performance pose significant design challenges.

To satisfy system-level requirements, this work adopts a two-tail dynamic comparator [16] architecture with a preamplifier stage, as shown in Figure 8. When the clock signal, CLKN, is high, the differential outputs of the preamplifier are reset to ground. As CLKN transitions low, the outputs begin charging toward a high voltage at a rate determined by the input differential voltage. This process converts the input voltage difference into a time-domain asymmetry, which is resolved by the dynamic latch to produce a digital output based on the relative timing of the input crossings.

To minimize the impact of flicker noise (1/f noise), the preamplifier employs a PMOS input pair. The comparator operates entirely dynamically, under CLKN control, consuming no static power. However, a residual inefficiency remains: after each comparison, once the latch has fired, the preamplifier output nodes continue charging fully to Vdd, resulting in unnecessary power consumption.

The proposed design mitigates this inefficiency by introducing a current-limiting transistor into the preamplifier stage (Figure 8). This device leverages the threshold voltage of a PMOS transistor (|Vthp|) to reduce the output voltage swing. As a result, the saturation voltage of the preamplifier’s output nodes is clamped at (Vdd − |Vthp|) instead of reaching the full Vdd rail. As illustrated in Figure 9, the waveforms illustrate the voltage rise of the comparator preamplifier in the improved and conventional structures under different common-mode voltages (Vcm). This reduction in voltage swing directly translates to lower overall power consumption in the comparator.

The preamplifier stage dominates power consumption. During each operational cycle, its differential output nodes are charged from 0 V to (Vdd − |Vthp|). Assuming a total load capacitance at each output node, the power consumed by the preamplifier can be estimated as follows: (3)$$P_{\mathrm{total}}=\frac{1}{T_{\mathrm{clk}}}\int \mathrm{Vdd}\left(I_{\mathrm{tail}}\right)\cdot dt=\frac{1}{T_{\mathrm{clk}}}\int_{0}^{\frac{T_{\mathrm{clk}}}{2}}\mathrm{Vdd}\left(I_{1}+I_{2}\right)\cdot dt$$
where $I_{1}$ and $I_{2}$ denote the currents in the two branches of the preamplifier stage. By substituting the capacitance model into Equation (Equation 3), we can derive the relationship between power consumption and the threshold voltage of the current-limiting transistor: (4)$$P_{\mathrm{total}}=\frac{2C}{T_{\mathrm{clk}}}\mathrm{Vdd}\left(\mathrm{Vdd}-\left|\mathrm{Vthp}\right|\right)$$

This finding implies that the power reduction depends on the ratio of the PMOS threshold to the supply voltage. The average power consumption of the comparator was measured to be 144.33 and 100.66 µW for Vcm voltages of 1 and 2 V, respectively. The overall power reduction is nearly 30%, as detailed in Table 1.

For input voltages exceeding 2 V, the current from the input pair decreases, preventing a further delay increase in the subsequent stage, as shown in Figure 10b. Simultaneously, the operating window of the current-limiting transistor narrows, reducing the magnitude of power reduction (Figure 10a). The current-limiting transistor introduces an average delay of 170 ps, which is considered negligible in a CIS system.

### 3.3. DAC Architecture

Common DAC architectures for charge-redistribution SAR ADCs include binary-weighted, bridge-capacitor, and split-capacitor types. While the bridge-capacitor DAC offers a compact layout, its inherent non-linearity poses a critical drawback for CIS applications. Furthermore, using a polysilicon capacitor—which exhibits high parasitic capacitance—as the bridging element leads to significant performance degradation. Consequently, a conventional binary-weighted DAC is implemented with polysilicon capacitors, offering the highest matching accuracy while maintaining area efficiency.

### 3.4. Phase-Adjustable SAR Logic

Conventional SAR logic modules are typically comprised of a dual-chain of D-flip-flops. For a single-ended DAC, this architecture demands a D-flip-flop count exceeding three times the ADC’s resolution. A 10-bit SAR ADC, for example, would necessitate at least 60 D-flip-flops for its fundamental control signals, resulting in a design that is both area-intensive and power-hungry. To address these limitations, some studies have proposed dynamic SAR logic [17,18] units that employ dynamic signaling for improved energy efficiency.

This section presents a design inspired by this approach, substituting the D-flip-flops with programmable non-overlapping clock generation units, such as Figure 11. Fed by a system clock, these units produce an N-phase non-overlapping clock signal, which is critical for ensuring sufficient settling time between each comparison in the SAR ADC. This signal then drives the DAC switches via simple buffers and inverters.

A key advantage of this architecture is its flexibility. By adjusting the number of clock units, additional clock phases can be dynamically inserted to create time slots for advanced processing. This feature is leveraged to implement a switchable noise-shaping function: enabling noise shaping increases the number of phases to improve accuracy, while disabling it allows the ADC to achieve higher sampling rates. Furthermore, the multi-phase clocking can be configured to control multi-phase sampling schemes, allowing the design to be optimized for either enhanced resolution or faster speed.

The proposed SAR logic module achieves this enhanced functionality with a significantly reduced transistor count. The core control logic is implemented using only 10 non-overlapping clock units, replacing the conventional 40 D-flip-flops.

Given that each clock unit comprises 12 transistors compared to 22 in a typical D-flip-flop, the overall transistor count is reduced by approximately 86%, resulting in a more compact layout and lower power dissipation.

To generate an N-phase non-overlapping control signal, the clock units are connected in a cascaded chain. Specifically, the output (OUT) of each unit is connected to the input (IN) of the subsequent stage. For all units except the one corresponding to the MSB, the DELAY input is also driven by the output of the preceding stage. A feedback loop is created by feeding the outputs of all stages into a NAND gate, with the gate’s output connected to the input of the first unit. This topology effectively forms a ring oscillator that produces the desired multi-phase clock. Figure 12 illustrates this implementation with a 7-phase non-overlapping clock chain.

Furthermore, the non-overlap duration between clock phases can be precisely controlled by adjusting the size of the buffers on the DELAY signal path within each unit. This adjustability allows the circuit to be tailored to accommodate diverse design requirements.

### 3.5. Noise Shaping Architecture

In recent years, NS SAR ADC has emerged as a prominent architecture [19]. This approach leverages techniques initially developed for sigma-delta converters to suppress many of the non-ideal factors inherent to the conventional SAR topology [20]. Despite this significant advantage, the practical implementation of NS-SAR ADCs presents distinct challenges. As illustrated in Figure 4, these challenges primarily lie in the intricate design of the loop filter and the precise configuration of the noise-shaping signal path.

Early approaches to noise shaping commonly employed multi-path comparators [21,22]. This technique integrates the residual voltage as a current within the comparator itself by incorporating additional signal paths. However, a significant drawback of this method is the increased noise from the multi-path comparator, which can diminish the effectiveness of the noise shaping. More recently, an alternative method that performs signal integration using capacitors has gained prominence [9,10]. This approach advantageously avoids the noise penalty associated with multi-path designs.

The present work adopts the architecture [9] depicted in Figure 13 to realize noise shaping. This structure utilizes switches to dynamically reconfigure the topology of an integration capacitor, thereby enabling the sampling and subsequent integration of the residual voltage onto the DAC.

However, parasitic capacitances influence the behavior of these noise-shaping capacitors. Specifically, for top-plate connections, only inter-circuit crosstalk capacitance contributes to minor charge loss, while the majority of the integrated charge is retained. In contrast, for bottom-plate connections, the parasitic capacitance to ground actively participates in the charge redistribution process following the DAC conversion phase. This participation introduces an additional loss in the integrator’s gain.

Consequently, this effect manifests in the system’s final noise transfer function (NTF) as a reduction in the residue voltage gain. Furthermore, it allows the common-mode voltage to introduce low-frequency harmonics, ultimately degrading the overall noise-shaping performance.

Figure 14 presents the timing diagram for the SAR logic. During the $\Phi_{1}$ phase, two integration capacitors (each C/2), storing the residual voltage from the preceding conversion cycle, are connected between the top plates of the differential DAC array and the comparator inputs. This configuration, which combines a series connection with differential operation, achieves a voltage gain of four for the residue.

Subsequently, once the SAR logic completes a full conversion, the $\Phi_{2}$ phase is initiated. In this phase, the voltage on the top plates of the DAC is transferred to the integration capacitors, which are now placed in parallel. This transfer is accomplished via charge redistribution.

### 3.6. Prototype Simulation

A preliminary behavioral model of the prototype device was implemented in Matlab Simulink, as illustrated in Figure 15.

The resulting quantization NTF is given by $\mathrm{NTF}=1-0.8z^{-1}$. A theoretical analysis of this NTF was performed and compared with that of an ideal first-order noise shaper. The results of this comparison are depicted in Figure 16.

To quantify the impact of parasitic capacitances on the NTF gain, we performed a parasitic extraction on the layout of the integration capacitor. Despite employing standard mitigation techniques, such as guard rings and a common-centroid layout, the post-layout simulation revealed that the parasitic capacitance on the bottom plate of the integrator reached approximately 20% of the integration capacitance itself. This gain degradation is attributed specifically to the bottom-plate parasitic capacitance, which actively participates in the charge redistribution process and thus reduces the integrator’s gain. This results in an overall gain reduction of about 10% in the noise-shaping loop.

This 10% gain reduction was subsequently introduced into our behavioral model to visualize its impact. As shown in the updated Figure 16, the resulting NTF curve demonstrates attenuated noise suppression compared to the theoretical curve, which aligns well with our measured results and highlights a key performance limitation in practice. It is worth noting that advanced techniques, such as optimized routing [23] or pre-charging schemes [24], could mitigate this issue. These methods, however, were not employed in this design. Given the limited area available for a column-level ADC, prioritizing architectural simplicity was a key design constraint, though these advanced techniques present a promising direction for future work.

The analog domain implementation utilizes a variable-gain, voltage-domain feedback loop comprising capacitors and an integrator. In this scheme, the residual voltage from the comparison in cycle N-1 is stored on the integration capacitor and subsequently added to the top-plate voltage of the DAC during cycle N via charge redistribution. This redistribution is realized through a configurable network of capacitors and switches. A key advantage of this approach is that, unlike conventional designs that use multi-channel comparators to perform a mixing function, it requires no modification to the comparator itself, thereby reducing its design complexity.

Designs that utilize multi-channel comparators for signal mixing typically require two or three parallel signal paths for first-order or second-order shaping, respectively. Mismatches between the amplification transistors in these paths inherently introduce more noise than a single-channel comparator. The proposed topology, which reconfigures the capacitor network using switches, successfully circumvents this issue.

## 4. Results and Discussion

### Measurements

A prototype of the proposed NS SAR ADC was implemented and taped out using the AMS C35B4OA 0.35µm technology. To evaluate its performance characteristics within a practical application context, the physical layout was designed with two parallel column ADC channels, as shown in Figure 17. This configuration is representative of a typical CIS column ADC’s architecture, ensuring that the subsequent performance evaluation accurately simulates the ADC’s behavior within its intended operational environment.

The prototype chip’s performance was evaluated under two distinct operational modes using a high-precision 3.3 V supply. The chip was driven by a differential, near-full-swing sine wave generated by a low-distortion signal source.

In standard SAR mode, the chip operated with a 24 MHz system clock and a 12-cycle conversion period, achieving a sampling rate of 2 M Sample/s. Under these conditions, a 23.3 kHz sinusoidal input (−0.83 dbFs) was applied, and the measured FFT spectrum is shown in Figure 18a.

Switching to noise-shaping mode, a 32 MHz clock with a 16-cycle conversion and an oversampling ratio (OSR) of 16 was utilized, which yielded an effective sampling rate of 125 kSa/s. In this configuration, a 6.57 kHz sinusoidal input was applied with a slightly reduced amplitude (−1.12 dBFS). The corresponding output FFT spectrum is presented in Figure 18b. Consistent with the theoretical noise transfer function (NTF) shown in Figure 16, the output spectrum (log version) presented in Figure 19 exhibits approximately 10 dB of noise suppression in the low-frequency region and a first-order noise-shaping slope of 20 dB/decade beyond the signal bandwidth.

To evaluate the ADC’s performance across different operating conditions, its dynamic and frequency-domain characteristics were measured by varying the input signal’s amplitude and frequency. The results are presented in Figure 20. Figure 20a plots the measured output spectrum as a function of input signal power, from which a Dynamic Range (DR) of 72.6 dB is determined. The frequency response of the ADC prototype is characterized in Figure 20b. The plot demonstrates a notably flat magnitude response across the low-to-mid-frequency range. This characteristic is highly suitable for CIS applications, which require high precision at low to medium operating speeds.

Despite prior efforts to reduce comparator power, its consumption still accounts for a significant portion of the total chip power, a challenge inherent to the 350 nm process with a 3.3 V supply. The power penalty from implementing noise shaping, however, is comparatively negligible. This overall power profile remains acceptable for the intended visible-light CIS applications.

The prototype was fabricated in AMS 0.35 µm technology, featuring two parallel column ADC channels, a configuration representative of typical CIS architecture. Test results indicate that noise shaping moderately improves the precision of the SAR ADC, which utilizes PIP capacitors for its capacitive DAC (CDAC). However, a notable discrepancy remains between measured and theoretical performance. The output spectra exhibit numerous spurious tones beyond expected harmonics, likely attributed to two factors: distortion from parasitic capacitance and capacitor mismatch [25] in the CDAC and spectral leakage from harmonics due to clock jitter [26,27]. Furthermore, the layout demonstrates that using a non-overlapping clock scheme for SAR logic provides tangible benefits in ADC-intensive systems such as CIS. A single, centralized control block can drive hundreds of ADCs, reducing the area of a single-column ADC by approximately 20–30%.

The performance comparison with previous works is listed in Table 2. To provide a comprehensive evaluation for CMOS image sensor (CIS) applications, the table encompasses both intrinsic ADC metrics and critical system-level performance indicators.

As detailed in Table 2, our work demonstrates a highly competitive NS SAR ADC tailored for high-resolution CIS applications. A key contribution of this work is achieving an excellent ENOB of 11.06 in a cost-effective 350 nm CMOS process. This level of precision is notably superior to other SAR ADCs, like the 8.35-bit ADC in, and is comparable to the performance of while supporting a full 1024 × 1024 pixel array at a standard video frame rate of 60 fps.

The primary advantage of our proposed architecture lies in its phase-adjustable noise-shaping technique, which is specifically designed to counteract the performance degradation caused by parasitic capacitances from poly-insulator-poly capacitors. This is a critical challenge in mature process nodes like 350 nm, where such parasitics are often significant. While the ΔΣ ADC in achieves a higher DR, our NS SAR architecture provides a more balanced solution for megapixel-resolution imaging systems, achieving a respectable 72.6 dB DR with lower design complexity, reduced power consumption, and a higher frame rate.

Furthermore, when evaluating the trade-offs, our design strikes a compelling balance between performance, power, and area. The power consumption of 125.7 µW per column is substantially lower than that of and, showcasing the energy efficiency of the SAR architecture. While reports lower power, its extremely low sampling rate is insufficient for the video-rate data conversion required by a megapixel sensor, which our work successfully enables. Therefore, this work validates that our proposed technique not only effectively mitigates a key hardware limitation but also enables the implementation of high-performance, high-resolution CIS for real-time imaging in mature and widely accessible semiconductor technologies.

## 5. Conclusions

This paper presents the design and characterization of a mode-switching column-level ADC tailored for CIS applications. The prototype, fabricated with the AMS C35B4OA 0.35 µm process, supports two operational modes: a high-speed SAR mode achieving 7.69-bit resolution at 2 MS/s while consuming 213.7 µW and a high-resolution noise-shaping mode delivering 11.06-bit resolution at 125 kS/s with a power consumption of 251.4 µW, whose power breakdown is detailed in Figure 21.

Performance analysis reveals parasitic capacitance in the PIP capacitor array as the primary limiting factor. To mitigate this bottleneck, the design incorporates adjustable phase control logic and a noise-shaping module, strategically trading sampling speed for improved in-band noise suppression. Despite residual noise artifacts from parasitics and jitter, measurement results confirm a substantial enhancement in spectral performance due to noise shaping.

Furthermore, the adjustable phase control logic provides the SAR ADC with significant operational flexibility within the CIS architecture. This flexibility enables switching between noise-shaping mode and high-speed mode, as well as controlling the sampling rate, resulting in an 86% reduction in control logic area. Specifically, at a sampling rate of 125 k Sample/s, the 1024 × 1024 pixel array in a double-column shared configuration achieves a frame rate of 60 fps in noise-shaping mode. In contrast, in high-speed mode, under the same configuration, the system can achieve 976 fps when the sampling rate is increased to 2 M Sample/s. In a two-column sharing configuration, the proposed ADC supports a 1024 × 1024 pixel array at a 60 Hz frame rate, demonstrating its suitability for high-resolution imaging applications.

## Figures and Tables

**Figure 1 sensors-25-06029-f001:**
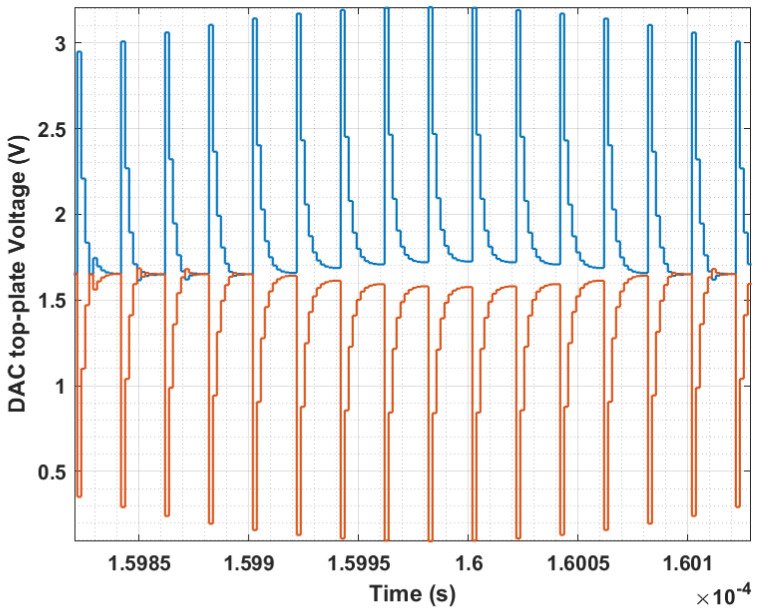
The DAC’s output range is limited by parasitic capacitance, which prevents a near-full-swing input signal from settling within 1 LSB at the top plate.

**Figure 2 sensors-25-06029-f002:**
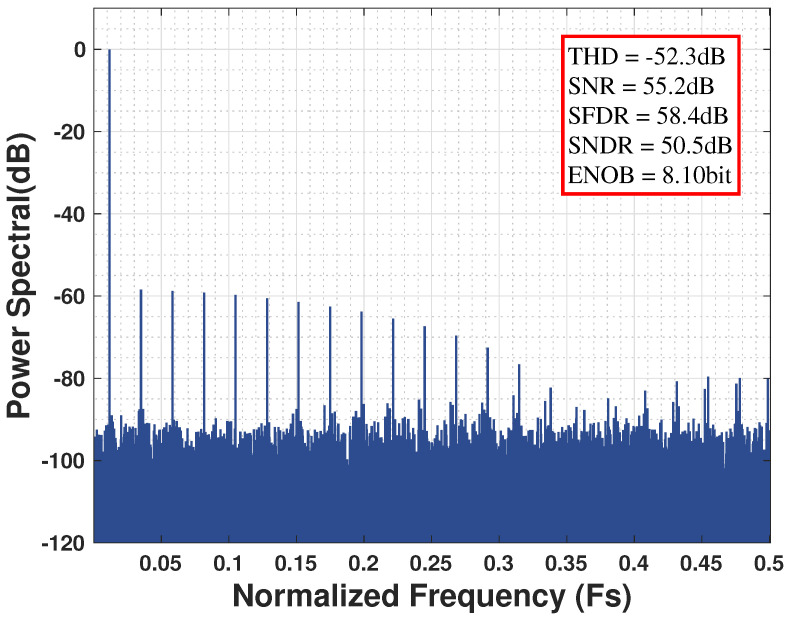
Harmonics generated by parasitic capacitance in the ADC output spectrum.

**Figure 3 sensors-25-06029-f003:**
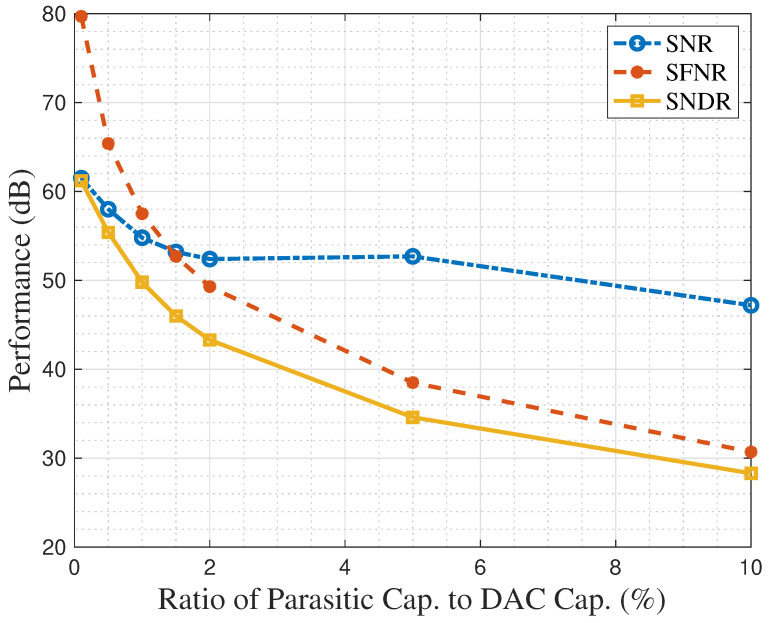
The performance metrics of the ADC degrade with increasing parasitic capacitance.

**Figure 4 sensors-25-06029-f004:**
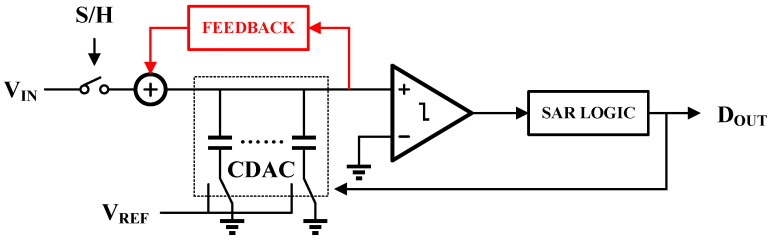
Block diagram of the proposed NS-SAR ADC. The conventional SAR core is shown in black, while the noise-shaping feedback loop is highlighted in red.

**Figure 5 sensors-25-06029-f005:**
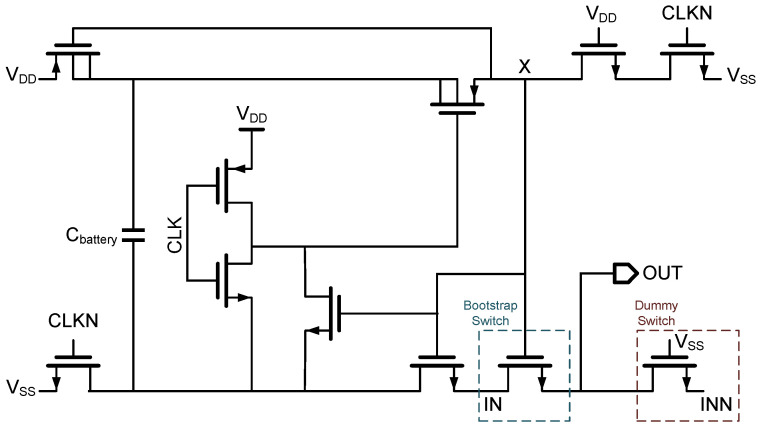
Schematic of the bootstrapped sample-and-hold switch with the proposed dummy switch for feedthrough cancellation. (Node X connects the gate and source to Cbattery, stabilizing the gate-source voltage and reducing input-induced nonlinearity.)

**Figure 6 sensors-25-06029-f006:**
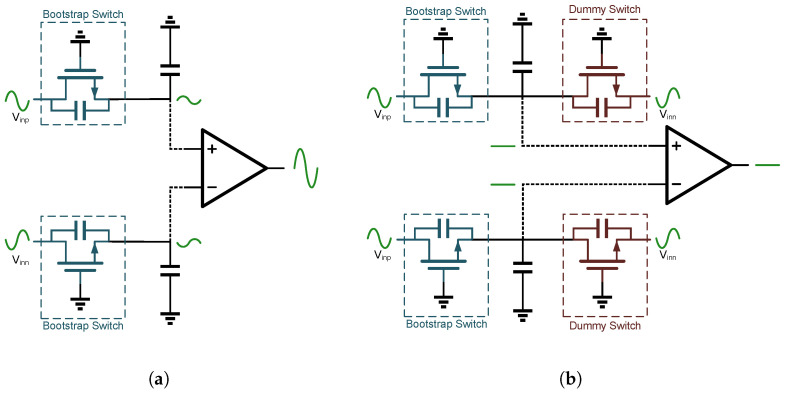
Bootstrapped switch with a dummy. (**a**) Conventional Bootstrap, (**b**) Proposed Bootstrap.

**Figure 7 sensors-25-06029-f007:**
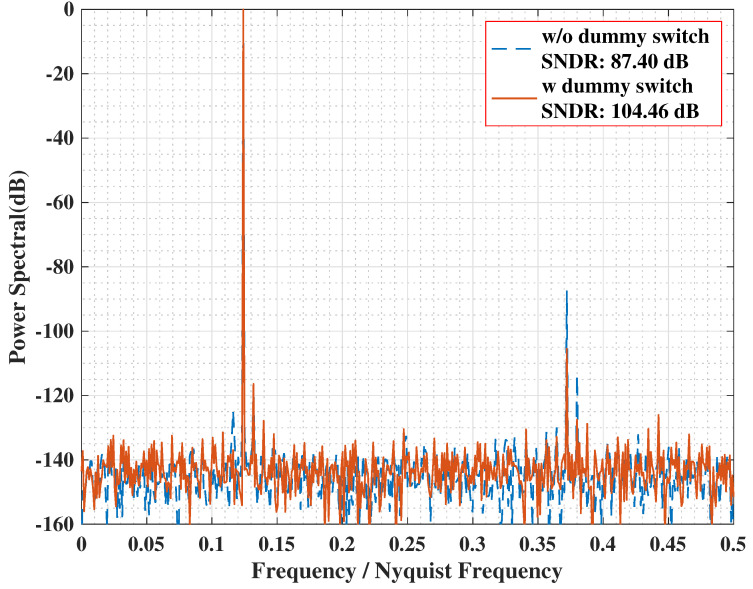
FFT spectrum of Bootstrap simulation.

**Figure 8 sensors-25-06029-f008:**
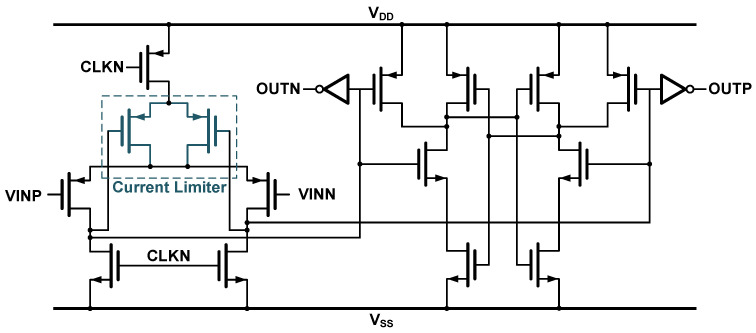
Low-power dynamic comparator schematic.

**Figure 9 sensors-25-06029-f009:**
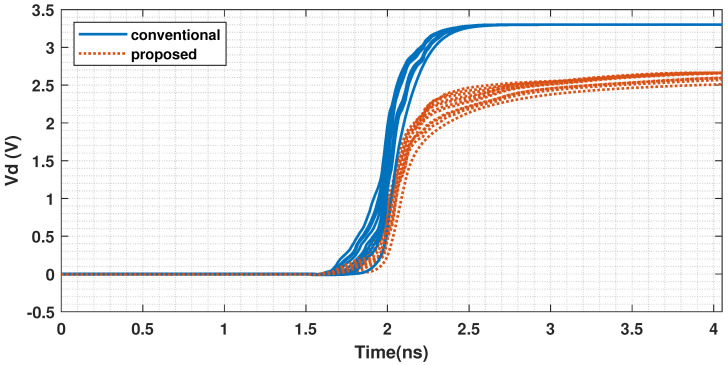
Simulated transient waveforms of the preamplifier’s output node for the conventional and proposed comparators, demonstrating the voltage swing reduction (Vcm from 0.5 V to 2.4 V).

**Figure 10 sensors-25-06029-f010:**
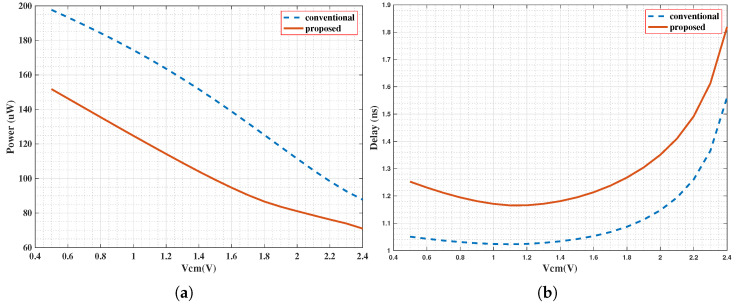
Comparator performance versus Vcm (Vdiff = 3 mV) of the proposed and conventional comparators. (**a**) Power consumption versus Vcm, (**b**) Delay versus Vcm.

**Figure 11 sensors-25-06029-f011:**
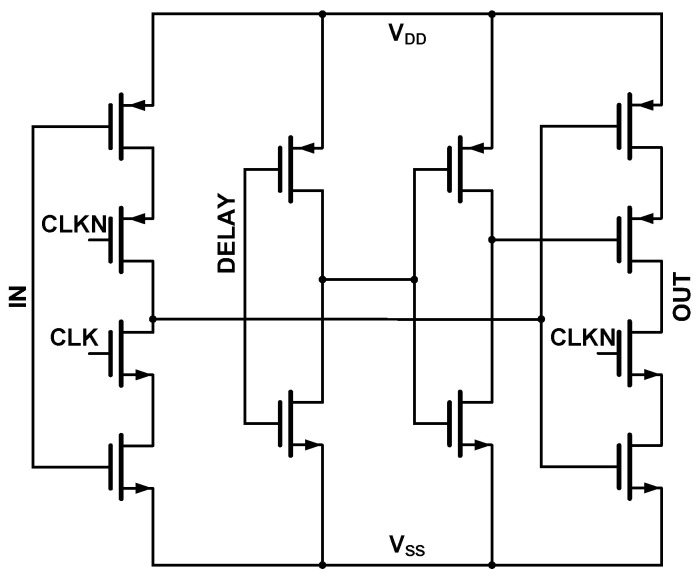
Schematic of the programmable non-overlapping clock generation cell.

**Figure 12 sensors-25-06029-f012:**
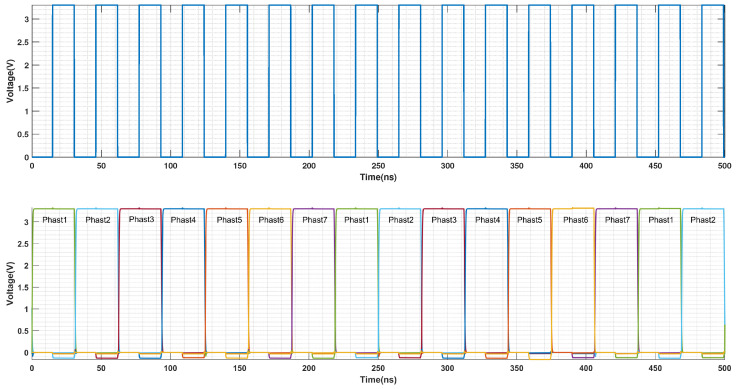
Timing diagram of SAR control logic (with 7-phase non-overlapping clocking).

**Figure 13 sensors-25-06029-f013:**
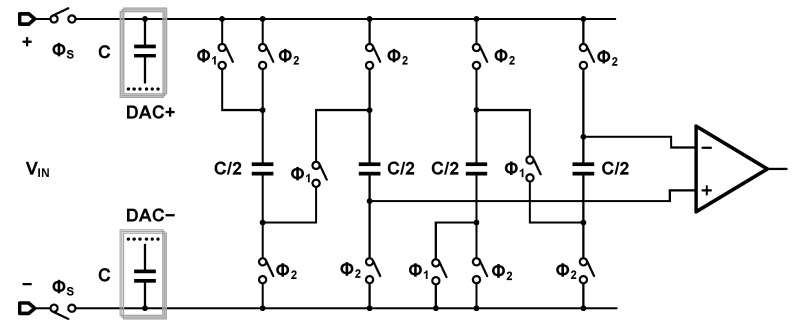
Noise-shaping integration capacitors.

**Figure 14 sensors-25-06029-f014:**
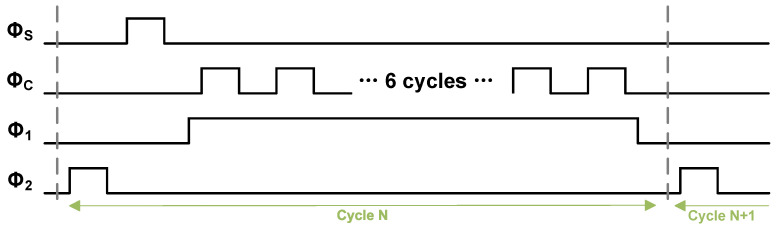
Timing diagram of control signals for noise-shaping SAR ADC.

**Figure 15 sensors-25-06029-f015:**
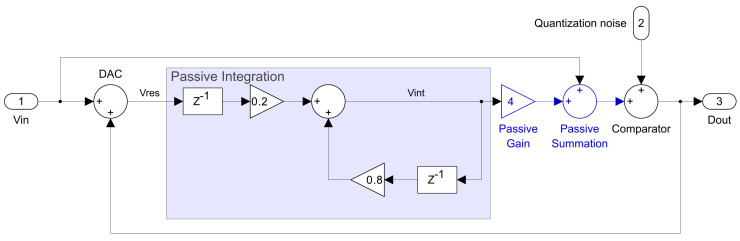
Noise-shaping SAR ADC signal flow diagram, implemented in Simulink.

**Figure 16 sensors-25-06029-f016:**
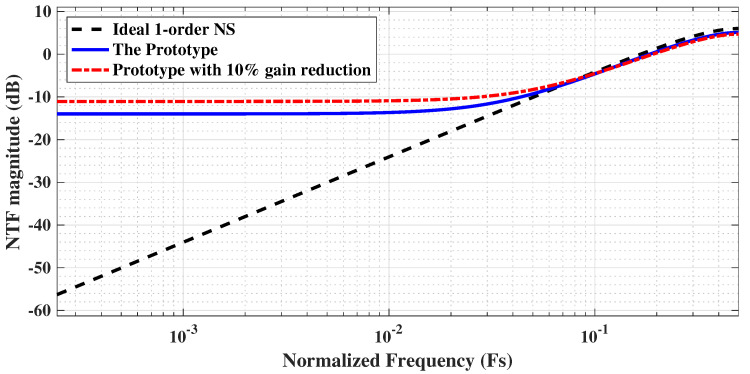
Prototype vs. ideal NTF of the 1-order noise shaping.

**Figure 17 sensors-25-06029-f017:**
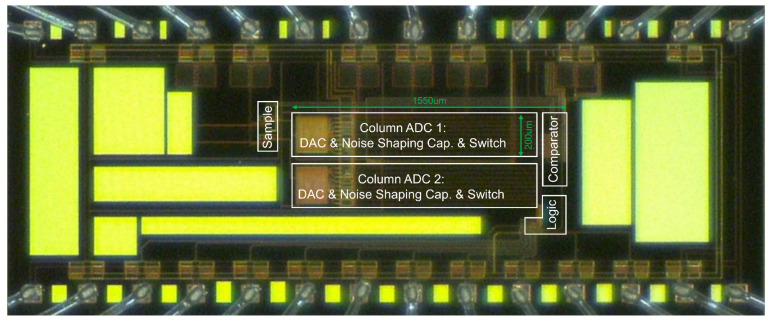
Die microphotograph.

**Figure 18 sensors-25-06029-f018:**
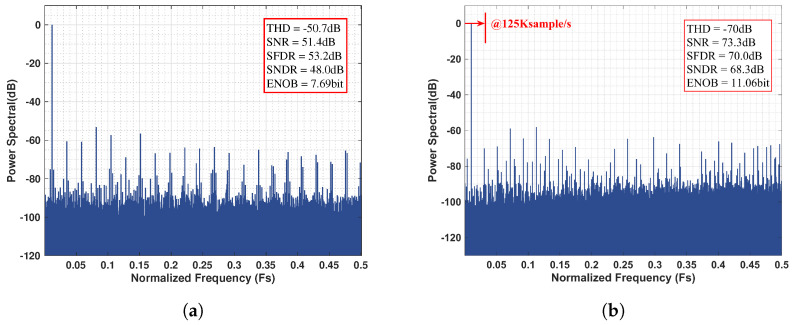
Comparison of simulated output spectra between practical and ideal noise-shaping SAR ADC models. (**a**) SAR ADC measured output spectral, (**b**) Noise-shaping SAR ADC measured output spectral.

**Figure 19 sensors-25-06029-f019:**
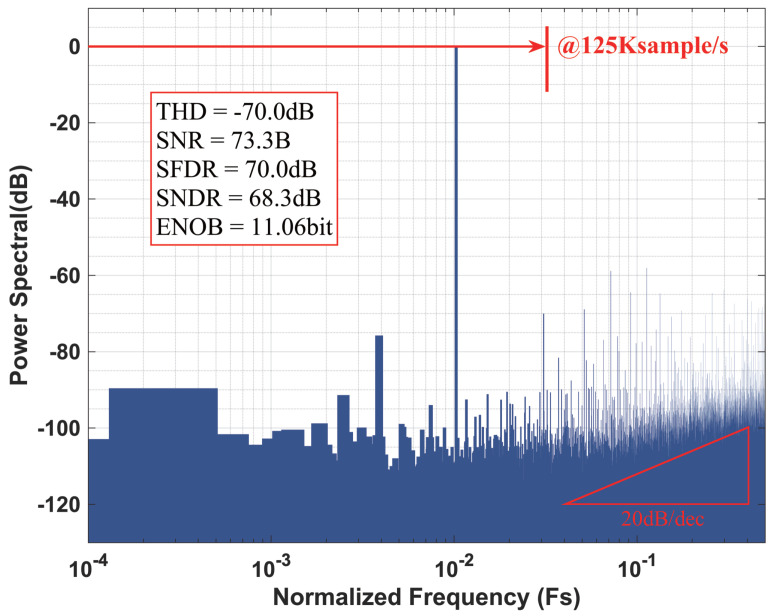
Noise-shaping SAR ADC measured output spectral (Log).

**Figure 20 sensors-25-06029-f020:**
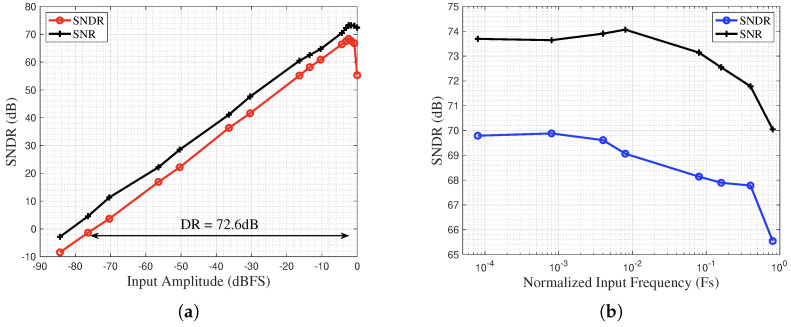
SNDR versus input signal amplitude and frequency for the NS SAR ADC. (**a**) SNDR vs. Input Signal Amplitude, (**b**) SNDR vs. Input Signal Frequency.

**Figure 21 sensors-25-06029-f021:**
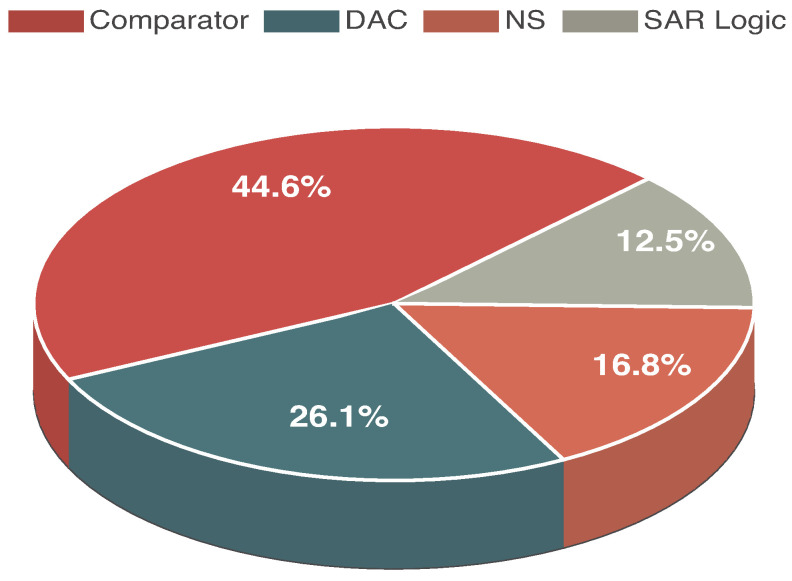
Power breakdown.

**Table 1 sensors-25-06029-t001:** Comparator power and delay performance of the 4T pixel across its operating range.

$V_{\mathrm{cm}}$ (V)	Conv. (µA)	Prop. (µA)	Conv. (ns)	Prop. (ns)
1.0	52.84	37.80	1.023	1.170
1.2	49.57	34.61	1.024	1.165
1.4	45.96	31.53	1.033	1.180
1.6	42.06	28.68	1.052	1.213
1.8	37.94	26.25	1.087	1.267
2.0	33.77	24.57	1.147	1.350

**Table 2 sensors-25-06029-t002:** Comparison with other published ADCs.

	Spivak [28]	Chen [29]	Lee [30]	Liu [31]	Brenna [32]	This Work
Process	350 nm	350 nm	130 nm	180 nm	350 nm	350 nm
Supply (V)	3.3	\	2.8/1.5	1.8	1.8	3.3
Pixel Array	32 × 32	800 × 600	640 × 480	\	\	1024 × 1024
ADC Type	Pixel	ΔΣ	TSSS	SAR	SAR	NS SAR
Frame Rate (fps)	\	50	320	\	\	60
Sampling Rate (MHz)	\	\	\	8.3	0.1	0.125
DR (dB)	58	84	75	\	\	72.6
NS Mode	No	Yes	No	No	No	Yes
ENOB (bit)	6.85	\	\	8.35	11.45	11.06
DNL (LSB)	\	−0.4/+0.8	−0.49/+1.34	−0.89/+0.98	−1/+1	−0.59/0.89
INL (LSB)	\	−3/+0.5	−2.47/+2.44	−1.8/+1.9	−2.6/+2.1	−1.58/1.36
Area (mm^2^)	\	0.024	0.0066	0.05	1.35	0.073
Power * (μW)	192.0	241.2	\	500	43.4	125.7

* The power here refers to the average power consumption of the ADC in each column of the pixel array.

## Data Availability

Data are contained within the article.

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
