# Peer review of "A Phase-Adjustable Noise-Shaping SAR ADC for Mitigating Parasitic Capacitance Effects from PIP Capacitors"

_sensors, 2025, doi:10.3390/s25196029_

Round 1

Reviewer 1 Report

Comments and Suggestions for Authors

General Comments

The topic is interesting, but the paper feels somewhat unorganized, and the reported performance is unfortunately not competitive. I think, with proper revision and measurements, the paper could be improved. Below are my detailed comments for the authors’ consideration:

  1. Section 2 – Analysis of Parasitic Effects
    While the issue of parasitic capacitance is important, the last two paragraphs of this section do not appear to be directly related to the main issue discussed. I recommend revising this part to improve clarity and ensure it remains focused on the parasitic effect problem.

  2. Section 3.5 – Capacitor Stacking
    Please clarify how parasitic capacitances specifically affect the capacitor stacking in this section. I think the explanation is incomplete and would benefit from more technical details.

  3. Figure 15
    The measured spectrum shows significant harmonic tones, which severely degrade the SFDR compared to the ideal case. The authors mentioned that these tones originate from sampling switches and the DAC. Could the authors clarify why such harmonics are not considered problematic?

  4. Figures 17–18 – Quality and Consistency
    The quality of Figures 17 and 18 is quite poor. Please improve the resolution and overall consistency of these figures (and the others). It is recommended to check how published ADC papers (e.g., JSSC/TCAS) present similar plots for reference.

  5. Power Breakdown
    A detailed power breakdown figure is missing from the paper. Including a breakdown of power consumption by block (e.g., SAR logic, DAC, reference, clocking, etc.) would provide valuable insights for readers and allow fairer comparison with prior works.

  6. Measurement Results
    The reported measurement results show quite weak performance. The SNR is 84 dB, while the SNDR is only 62.4 dB, a gap of more than 20 dB. This indicates that distortion severely limits performance (as shown in SFDR measurement). Could this degradation be due to input amplitude saturation?

    In addition, most noise-shaping SAR ADC papers include dynamic range (DR) plots, showing SNDR/SNR versus input amplitude. Sometimes even versus input frequency. 

    Lastly, a table comparing the proposed ADC with other works can be included. Such a table is important to highlight the competitiveness of the work.

    The authors mentioned tin the paper that the performance degradation could be due to implementation issues such as parasitic capacitances, clock jitter, or capacitor mismatch. However, it would be helpful if the authors could provide justification or supporting evidence for this claim, such as additional simulations or analysis to quantify the impact of these factors. Furthermore, if the degradation is indeed due to implementation issues, it is unclear why these factors were not considered or mitigated prior to chip fabrication.

    From a system perspective, is the observed performance acceptable for the intended application? Addressing these points would strengthen the technical contribution of the paper.

    I also wonder if the authors implemented DWA or MES technique?

Author Response

Thank you for your review comments. I have summarized my responses in the attachment "MDPI Reply to Reviewer1.docx"; please check.

Reviewer 2 Report

Comments and Suggestions for Authors

Overall, the authors propose a new ADC architecture and improve SNR. The fabricated chip also shows improved SNR.

The following is a list of minor comments:

・Please indicate what FB in Figure 2 stands for in the caption. It is probably FB (Feedback).

  • Please indicate where the MOS in the lower left of Figure 3 is connected. Also, please include an explanation of node X, as it is not mentioned in the text.

  • Vcm may be common knowledge, but please include it in the text for clarity. Is it the common-mode voltage?

  • Figures 15, 17, and 18 are black and white, so it would be better to color them. Additionally, it may be better to change not only the colors but also the line types (e.g., dashes) in other figures.

・Regarding Chapter 4, please include a table detailing the specifications of the ADC you created. Indicate the bit count, DNL, INL, and area (size) of each ADC. Also, add a table comparing it with other papers on ADCs for image sensors. Also mention the errors for each chip and each ADC. 

・Regarding lines 310–311. This should be included in Chapter 4. Include this in Chapter 4 and compare it with other studies.

・Regarding line 322. In the figure16 shows that each ADC is fabricated with a width of 200 µm, but the authors mention that two pixels are shared. However, with this width, each pixel becomes 100 µm, which seems too large. Please confirm whether this number is correct. Additionally, regarding the frame rate, please clearly indicate whether this is in high-speed mode or noise-shaping mode. Also, please specify the FPS in high-speed mode and the FPS in noise-shaping mode.

Author Response

Thank you for your review comments. I have summarized my responses in the attachment "MDPI Reply to Reviewer2.docx"; please check.

Round 2

Reviewer 1 Report

Comments and Suggestions for Authors

Thank you for addressing my concerns from the previous round of review. I have few more comments and suggestions for improvement:

  1. References Update
    I recommend replacing reference [2] with the IEEE OJ-SSCS 2024 paper “SAR-Assisted Energy-Efficient Hybrid ADCs” (from the same research group), which is a more up-to-date and extended counterpart. This is preferable because the Journal of Semiconductor Engineering may be less familiar or less widely recognized among readers in the field.

  2. Parasitic Capacitance on Stacked Capacitors
    The manuscript now explains the effect of parasitic capacitances on stacked capacitors. Since this is a real issue that can significantly degrade performance, the paper could go further for completeness. In particular, the authors could clarify what specific design choices or mitigation techniques they employed (or chose not to employ) and provide justification. Additional references to prior works that addressed similar parasitic issues would also be helpful. Including these details would strengthen the paper, especially since noise-shaping SAR is one of the main focuses of the work.

  3. Figures 18(b) and 19 Consistency
    Figures 18(b) and 19 appear essentially the same (linear vs log scale), but there may be an inconsistency in the line marker for 125 kSamples/s. In Figure 18(b), the line seems to extend only up to the third vertical dashed line, placing the large harmonic tone outside the signal band. In Figure 19, the same (I assume) harmonic tone appears within the signal band. Please clarify this discrepancy, and revise if necessary.

  4. Minor Point – Input Signal Representation
    Instead of labeling the input as “DC” and “AC,” use dBFS notation. This is the standard for specifying input information in ADC performance plots and improves comparability with other works. Please refer to other ADC published works as an example.

  5. Missing SNR Data Points
    Add SNR data points (in addition to SNDR) in both the dynamic range plot and the input frequency sweep plot, especially since the main FFT plot show considerable harmonic tones. Additional measurement points near the MSA region in the dynamic range plot is also needed.

  6. Figure 3: Additional Details
    Include more quantitative discussion regarding Figure 3. For example, specify how much degradation the ADC experiences as the parameter varies from 1.5% to 10%. One possible phrasing: “The dominant source of performance degradation transitions from thermal noise to harmonic distortion, where SFDR is degraded by xx dB.”

  7. Performance Comparison Table Commentary
    The performance comparison table requires accompanying discussion. Simply presenting the table without interpretation adds little value. The authors should highlight insights such as the advantages, limitations, or trade-offs of the proposed ADC relative to the other works and explain how the table supports the paper’s main contributions.

  8. Minor Point – Power Breakdown Chart
    Consider using a pie chart format.

Author Response

Thank you for your review comments. I have summarized my responses in the attachment "MDPI Reply2 to Reviewer1.docx"; please check.
